# Transcatheter Mitral Valve-in-Valve Implantation with the Balloon-Expandable Myval Device

**DOI:** 10.3390/jcm11175210

**Published:** 2022-09-02

**Authors:** Sara Blasco-Turrión, Ana Serrador-Frutos, John Jose, Gunasekaran Sengotuvelu, Ashok Seth, Victor G. Aldana, Juan Pablo Sánchez-Luna, Jose Carlos Gonzalez-Gutiérrez, Mario García-Gómez, Javier Gómez-Herrero, Cristhian Aristizabal, J. Alberto San Román, Ignacio J. Amat-Santos

**Affiliations:** 1Cardiology Department, Hospital Clínico Universitario de Valladolid, 47003 Valladolid, Spain; 2Centro de Investigación en Red—Enfermedades Cardiovasculares, 28029 Madrid, Spain; 3Cardiology Unit 2, Christian Medical College Hospital Vellore, Vellore 632004, India; 4Department of Cardiology, Apollo Hospitals, Chennai 600006, India; 5Cardiology Department, Fortis Escorts Heart Institute, New Delhi 110025, India; 6Cardiology Department, Clinica Medicadiz, Ibagué 730006, Colombia

**Keywords:** valve-in-valve, mitral bioprosthesis, TMVR, Myval, Sapien

## Abstract

**Background:** The vast majority of transcatheter valve-in-valve (ViV) mitral procedures have been reported with the SAPIEN family. We aimed to report the preliminary experience with the Myval balloon-expandable device in this setting. **Methods:** Multicenter retrospective study of high-risk surgical patients with mitral bioprosthesis degeneration undergoing transcatheter ViV implantation with Myval device. **Results:** A total of 11 patients from five institutions were gathered between 2019 and 2022 (age 68 ± 7.8, 63% women). The peak and mean transvalvular gradients were 27 ± 5 mmHg and 14.7 ± 2.3 mmHg, respectively, and the predicted neo-left ventricular outflow tract (neo-LVOT) area was 183.4 ± 56 mm^2^ (range: 171 to 221 mm^2^). The procedures were performed via transfemoral access in all cases (through echocardiography-guided transeptal puncture (81.8% transesophageal, 11.2% intracardiac)). Technical success was achieved in all cases, with no significant residual mitral stenosis in any of them (peak 7.2 ± 2.7 and mean gradient 3.4 ± 1.7 mmHg) and no complications during the procedure. There were no data of LVOT obstruction, migration, or paravalvular leak in any case. Mean hospital stay was 3 days, with one major vascular complication and no stroke. At 6-month follow-up, there was one case with suboptimal anticoagulation presenting an increase in the transmitral gradients (mean 15 mmHg) that normalized after optimization of the anticoagulation, but no other relevant events. **Conclusions:** Transseptal ViV mitral implantation with the balloon-expandable Myval device was feasible and safe avoiding redo surgery in high-risk patients with bioprosthesis degeneration.

## 1. Introduction

Up to 35% of the patients harboring a mitral bioprosthesis will need a redo surgery within the first 10 years after the prior procedure [1]. However, redo mitral valve surgery is a high-risk procedure with a mortality rate that grows with each reintervention, reaching 15% for a second surgery and 40% after a fourth redo mitral intervention [2]. Transcatheter mitral valve replacement (TMVR) using balloon-expandable aortic transcatheter heart valves was proposed over a decade ago as an alternative to open-heart surgery for patients with severe mitral valve disease due to degenerated bioprostheses or failed surgical repair with annuloplasty rings, as well as in selected patients with native mitral valve disease and severe mitral annular calcification who are not eligible for conventional surgery. Despite the success and rapid evolution of transcatheter aortic valve-in-valve (ViV) replacement over the years, TMVR has not flourished as much and there are no clinical trials comparing TVMR with redo surgery. 

Although the evidence is mostly based on clinical registries with the Edwards SAPIEN prosthesis family (Edward Lifesciences, Irvine, CA, USA), recently some preliminary experience has been reported with the balloon-expandable Myval system (Meril Life Sciences Pvt Ltd., Vapi, India) [3,4,5]. The Myval device obtained the European Community (CE) mark in 2018 following the Myval-1 Study for the treatment of aortic stenosis, and thereafter aortic valve-in-valve (ViV) procedures were performed successfully [6]. The structure of the valve is similar to the SAPIEN 3 platform, but has a greater variety of sizes available (conventional sizes: 20, 23, 26, and 29; intermediate sizes: 21.5, 24.5, and 27.5; and extra-large sizes: 30.5 and 32 mm) and a lower entry profile (14 French), which could both be particularly useful for mitral ViV procedure [7].

Therefore, we aimed to describe the safety and feasibility of mitral ViV procedures with the Myval device in a short series of consecutive patients.

## 2. Methods

Multicenter and retrospective registry of patients with mitral bioprosthesis degeneration underwent ViV TMVR with the balloon-expandable Myval device after heart team approval. Participating centers used standardized definitions to collect clinical information, including demographic characteristics and procedural details. The study was approved by local ethics committees, and all patients provided written informed consent for the study.

### 2.1. Study Endpoints

The primary safety endpoint of the study was to assess procedural technical success, defined by the Mitral Valve Academic Research Consortium (MVARC) criteria at exit from the catheterization laboratory as: patient alive with successful access; delivery and retrieval of the device delivery system; successful deployment and correct position of the first intended device; and freedom from emergency surgery or re-intervention associated with the device or access procedure. 

The secondary endpoints were the absence of significant residual mitral stenosis (defined as immediate postprocedural mean gradient ≥ 10 mmHg) and significant residual mitral regurgitation (defined as regurgitation ≥ moderate). 

### 2.2. Imaging Analysis

Transthoracic echocardiograms (TTE) were obtained at baseline, at 30-day and at 6-month follow-up, and the measured parameters followed the recommendations from the European and the American guidelines [8]. The hemodynamic performance of each THV was assessed by the degree of mitral regurgitation, residual transvalvular gradient, and estimated area (MVA).

Multidetector computed tomography (MDCT) exams were performed according to the guidelines of the Society of Cardiovascular Computed Tomography [9]. The following derived parameters were calculated: mitral prosthesis inner annulus (maximal, minimal, and mean diameters), area, and perimeter.

### 2.3. Data Collection and Statistical Analysis

All baseline, procedural, 30-day, and 6-month follow-up data from the study population were retrospectively collected in a dedicated database at each participating institution. Categorical variables are presented as frequencies and percentages. Continuous variables are expressed as mean (±standard deviation) or median (25th–75th interquartile range). All analyses were performed using R software, version 3.6.1 (R Project for Statistical Computing, Lucent Technology, Reston, VA, USA).

## 3. Results

### 3.1. Study Population

A total of 11 patients from five different centers underwent Myval transcatheter ViV implantation between 2019 and 2022. No mitral valve-in-ring or valve-in-mitral annular calcification were included in this registry. The mean patient age was 68 ± 7.8 years, 63% were women, and the main indication for the procedure was mitral bioprosthesis stenosis. All patients were considered of high risk for surgical valve replacement after the Heart Team evaluation, and therefore eligible for ViV procedure. Baseline demographic and previous bioprosthesis characteristics are presented in Table 1. The peak and mean transvalvular gradients were 27 ± 5 and 14.7 ± 2.3 mmHg, respectively, and the predicted neo-left ventricular outflow tract (neo-LVOT) area was 183.4 ± 56 mm^2^ (range: 171 to 221 mm^2^). 

### 3.2. Procedural Outcomes

All patients underwent transcatheter ViV implantation with the balloon-expandable Myval device through transseptal access by a bilateral common femoral vein access. Transeptal puncture was echocardiographically guided in all cases (81.8% transesophageal, 11.2% intracardiac) (Appendix A), and 12- or 14-mm balloons were used for septal dilation in five and six cases, respectively. The baseline and post-procedural echocardiographic parameters are listed in Table 2. 

The size of the Myval device and the balloon used for valvuloplasty, when needed, were selected based on the true internal diameter (ID) of the previous surgical bioprosthesis from MDCT measurements; there were no specific recommendation in the “valve-in-valve” app [10]. The Myval size ranged from 23 mm to 29 mm, and intermediate sizes were required in 54% of the patients. In 90% of the patients, the implanted valve was oversized, whilst in one patient a smaller valve was used with higher inflation volume because of severe tissue ingrowth. Balloon valvuloplasty was performed under rapid ventricular pacing in four patients, and postdilatation was also required in these same four patients (Figure 1) [11]. There was no need for iatrogenic atrial septal defect closure in any of the patients. Technical success was achieved in all cases, with no significant residual mitral stenosis in any of them (peak and mean gradient of 7.2 ± 2.7 and 3.4 ± 1.7 mmHg, respectively).

The primary safety endpoint of technical success defined by MVARC criteria at exit from the catheterization laboratory was achieved in all the cases. The secondary endpoint was also achieved in 100% of the cases as the echocardiographic parameters showed an average reduction in mean gradients of 10.7 ± 4 mmHg with no cases postprocedural mean gradient above 5 mmHg or significant paravalvular residual regurgitation in any of the patients. There were no data of LVOT obstruction, migration, or paravalvular leak in any case, and no perforation or pericardial effusion. There was no need for a second TMVR implantation during index procedure in any of the patients, no intraprocedural mortality nor conversion to open heart surgery. Mean hospital stay was 3 days, with one major vascular complication and no stroke. One patient suffered from major bleeding during the procedure with no further repercussion.

### 3.3. Results at Follow Up

At 6-month follow up, no major complications were reported, including stroke, device embolization, left ventricular outflow tract (LVOT) obstruction and new paravalvular leak. All patients were receiving oral anticoagulation at baseline (9 vitamin K inhibitors and 2 apixaban) and same regime was maintained after discharge. Ten patients did not require new hospitalizations, but one presented a heart failure re-admission and an increase in the transmitral mean gradient (15 mmHg) was detected; the range of anticoagulation had been suboptimal and gradients normalized after optimization of the anticoagulation (Appendix A). No other events were recorded. Peak and mean transmitral gradients were 7.5 ± 2.9 and 3.6 ± 2.0 mmHg 

## 4. Discussion

In this preliminary experience with the Myval device for ViV procedures in mitral position, the main findings were: (1) The procedure was safe with technical success in all patients; in particular, septal crossing was feasible in every case after predilation without any valve dislodgement or significant septal tear. (2) Optimal hemodynamics were obtained including adequate transprosthetic gradients and lack of neo-LVOT obstruction by following the same recommendations as for alternative balloon-expandable devices [12]. The availability of intermediate and extra-large sizes with the Myval device might allow a more precise sizing, but since no official recommendation existed and the device was not included in the ViV app [10], a valve sizing chart was elaborated following the surgical prosthesis inner area (based on theoretical internal diameter or true measured area) and the Myval device areas at nominal pressure (Table 3). (3) The third finding of this research confirms that optimal medical therapy following mitral ViV procedures remains unclear, while the risk of leaflets hypoattenuation or clinical/subclinical leaflet thrombosis remains higher than for aortic ViV procedures (9.1% in our research). Given the growing number of this type of procedures, deeper investigation is crucial to determine this relevant aspect.

### 4.1. Surgical Mitral Biosprosthesis: Degeneration and Management

Over the last decade, a growing concern regarding mitral bioprosthesis degeneration has arisen due to the aging of the population and the increasing use of bioprosthesis over mechanical heart valves. In the mitral position, bioprosthetic valve degeneration tends to occur faster than in the aortic position. In fact, at 10 years, surgical valve degeneration requiring reintervention reaches 20–30%, and at 15 years it ranges from 60 to 80% depending on the bioprosthetic type [11,12,13]. The high mortality and complication rate associated with redo open heart surgery, even in younger patients [14], has promoted the development of a less invasive approach (obtaining the FDA approval for percutaneous mitral ViV procedures in high-risk patients in 2017, and a IIb recommendation in the latest ESC/EACTS Guidelines for the management of valvular heart disease [11,15]). However, for the success of such procedures, knowing the previous surgical bioprosthesis and understanding the preceding mitral valve surgery is key. Although all surgical mitral valves are stented, there are differences in stent radiopacity that can make the positioning more challenging. In addition, the level at which leaflets are sutured inside the stent can play an important role in the risk of LVOT obstruction, as does whether the anterior mitral leaflet has been resected or not. Whether the interatrial septum has been previously crossed, closed or patched is important for the transseptal access. For all of these aspects, pre-procedural planning with imaging techniques is mandatory; MDCT allows determination of the true ID of the bioprosthesis (that can vary from what was reported by the manufacturer), but also helps to plan the ideal site for transseptal puncture (favoring a more inferior puncture over the superior and posterior puncture performed for edge-to-edge procedures) and the best fluoroscopy angles for THV deployment. In addition, simulation of a virtual valve allows estimation of the risk of LVOT obstruction during the ViV procedure not only by the simulated area of the neo-LVOT, but also by the analysis of several predictors of such complication including the aortomitral-annular angle, which is the angle between the annular planes of these two valves (if the angle is obtuse, there may be a higher risk of obstruction, as the struts of the prosthesis will encroach on the LVOT), the degree of septal hypertrophy, the left ventricle size, and device protrusion and flaring into the LVOT (Figure 2). An expected neo-LVOT area below 1.7 cm^2^ has high sensitivity and specificity for post-procedural LVOT obstruction [11] and, thus, it should be prevented by alcohol septal ablation (at least 4 weeks prior to the valve-in-valve procedure) or the electrical laceration of the anterior mitral leaflet (LAMPOON). However, none of these two rescue strategies have yet been approved by the FDA in this context [5,11].

### 4.2. Valve-in-Valve Mitral Procedures: Scientific Evidence

No clinical trial comparing ViV TMVR and open-heart surgery has been carried out, but several studies have reported promising results in terms of survival and clinical outcomes (therefore arising as the preferred alternative for patients with a degenerated bioprosthesis, regardless of age and baseline surgical risk [16,17]). ViV therapies are not only used for patients with degenerated bioprosthetic mitral valves, but also in failing annuloplasty rings (valve-in-ring) and selected cases of native valve disease with severe mitral annular calcification who are not eligible for conventional surgery (valve-in-MAC). The clear advantage in ViV mitral procedures is that the valve frame—circular and often radiopaque—offers an optimal anchoring point for the transcatheter heart valve. 

Several case reports, two large registries and a clinical trial have been published regarding mitral ViV. Firstly, the TVT registry, which included patients undergoing ViV TVMR from 2006 to 2021, reported the 1-year experience with 1529 SAPIEN 3 valves (86% via transseptal access) with high technical success (96.8%), 30-day and 1-year survival (93.5% and 82%, respectively) and significant improvement in heart failure symptoms and quality of life at 30-day that were maintained at 1-year follow-up [17]. Soon after, the VIVID registry reported their 4-year experience with 1857 mitral ViV replacement (mostly performed with the SAPIEN 3 valve and transapical access), with comparable 30-day and 1-year survival rates (93.5% and 86.2%, respectively) and a lower 4-year survival (62.5%) mostly related to significant post-procedural mitral regurgitation [16]. Lastly, the MITRAL trial recently published the 1-year outcomes of 61 prospectively enrolled transseptal mitral ViV and valve-in-ring patients confirming the promising results regarding clinical status (89.2% of the patients in NYHA class I or II) and survival rate (96.7%) at 1-year follow-up [18].

### 4.3. Valve-in-Valve Mitral Procedures: Approach, Devices, and Risks

To date, the balloon-expandable SAPIEN 3 transcatheter heart valve is the only one approved for ViV procedures in both aortic and mitral position, and some cases using the mechanically expanded Lotus valve and the self-expandable J System for ViV TVMR have been reported with good results in terms of technical success and clinical outcomes [1]. However, no data regarding other balloon-expandable devices have been published in this context. Lotus prosthesis was proposed as a safer option for ViV TVMR, owing to its complete repositionability and retrievability until the time of final release, also favored by its lower stent height (19 mm) compared to SAPIEN 3 valves (15.5–22.5 mm) [19]. However, it required a transapical approach, which is a higher-risk approach, and the device is currently off the market.

The Myval valve (Meril Life Sciences, Vapi, India), which obtained the CE mark in 2018 following the Myval-1 Study, is a next-generation balloon-expandable valve with a similar structure to the SAPIEN 3 valve, made from bovine pericardial tissue mounted in a nickel-cobalt frame that has been designed with some novelties such as an anti-calcification coating to reduce valve degeneration and an external polyethylene terephthalate (PET) buffing to avoid para-valvular leak. However, one of its main and encouraging advantages is its wider range of sizes available and its low-profile access, as all diameters are compatible with a 14 Fr sheath, reducing potential vascular complications [20]. This adequate profile—better than SAPIEN 3 for the larger devices—might also be advantageous for septal crossing. In our research, no significant residual interatrial shunt was detected in any case, and no impact in the right chambers was detected up to 6-months follow-up. Despite the reduced sample, this compares favorably to what has been reported with alternative devices (3.3–7.6% need for closure of atrial septal defect) [17,21]. In addition, availability of intermediate and extra-large sizes might represent an advantage, because choosing the correct size and degree of oversizing is crucial to achieve optimal results. An undersized valve may result in high risk of malposition, embolization or atrial migration. To prevent device migration or embolization, an oversizing strategy is preferred in ViV TMVR. However, excessive oversizing may lead to the distortion of the transcatheter valve and under-expansion of the device—which could increase the risk of device thrombosis, leaflet pin-wheeling, and premature degeneration [20]. In our study, more than half of the patients needed an intermediate-size valve; still, a close surveillance of gradients progression is recommended. In our cohort, cases 10 and 11 presented adequate mean gradients immediately post-procedure, but progressed to 7 and 7.6 mmHg, respectively, 72 h later. In this sense, although a residual transmitral mean gradient > 5 mmHg has been considered significant according to the MVARC, no association with poorer outcomes was detected in the VIVID Registry, (in contrast to what has been reported for the edge-to-edge mitral valve repair). In fact, the threshold for worse prognosis was set in a postprocedural mean gradient ≥ 10 mmHg, as it aroused as an independent predictor for mitral valve reintervention and worse heart failure symptoms. A more ventricular position of the valve is recommended for the ViV TMVR to avoid significant residual mitral gradients, and better gradients were obtained in our sample when a flail of the valve was induced by a second more ventricular balloon inflation (see Appendix A).

### 4.4. Unresolved Issues in Valve-in-Valve Mitral Procedures

There are certain concerns regarding the risk of valve thrombosis after transcatheter valves [22]. Although it is known that the risk of this complication is higher following ViV procedures than for native valves treated percutaneously, there is scarce information about the risk of valve thrombosis after TMVR. It is well known that dedicated devices for TMVR have presented this complication, including the Fortis valve (which temporarily halted the program due to valve thrombosis [23]), or the Tendyne valve [24]. Despite the extended use of balloon-expandable devices for mitral ViV, very little details about this complication or the antithrombotic regime are known. When identified, the treatment of valve thrombosis by intensification of the anticoagulation usually can solve the problem, but no study has assessed yet the risk and efficacy of adding antiplatelet therapy to the anticoagulation in such patients. Despite the good technical results and clinical outcomes reported so far, including the case reported in our investigation with the Myval device, the durability of transcatheter valves in the mitral position remains unknown and, as already said, the VIVID Registry suggests a significant increase in both mean gradients and residual mitral regurgitation from the first year of follow-up. Thus, the long-term outcome of ViV TMVR warrant further study. 

### 4.5. Limitations

This study has the inherent limitations of an observational study with limited independent adjudication of adverse events and potential underreporting of adverse events. Additionally, there was no independent echocardiographic central laboratory. 

## 5. Conclusions

The use of Myval device for percutaneous treatment of surgical mitral bioprosthesis degeneration through transeptal access was feasible and safe, and represents the first alternative to the balloon-expandable SAPIEN valves. Technical success, hemodynamics and early outcomes were optimal, but further studies with long-term follow-up are needed to confirm our results in this setting.

## Figures and Tables

**Figure 1 jcm-11-05210-f001:**
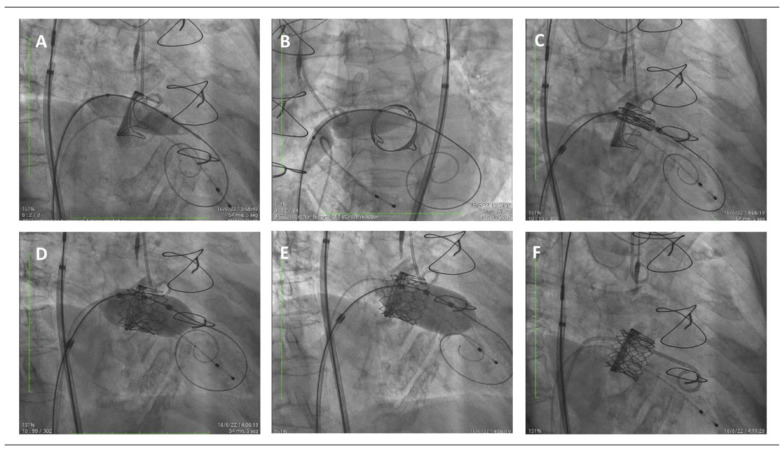
Step-by-step mitral valve-in-valve procedure, fluoroscopic images. (**A**) Balloon valvuloplasty with a 14 mm balloon over the Safari wire advanced through the interatrial septum and across the surgical mitral bioprosthetic valve. (**B**) Balloon dilation of the interatrial septum with the same 14 mm balloon. (**C**) Positioning of the transcatheter Myval device within the surgical bioprosthetic frame. (**D**) Deployment of the transcatheter valve inside the previous surgical bioprosthetic mitral valve. (**E**) Postdilatation with the same balloon slightly advanced towards the left ventricle aiming to provide a flail and preclude from atrial migration. (**F**) Final result.

**Figure 2 jcm-11-05210-f002:**
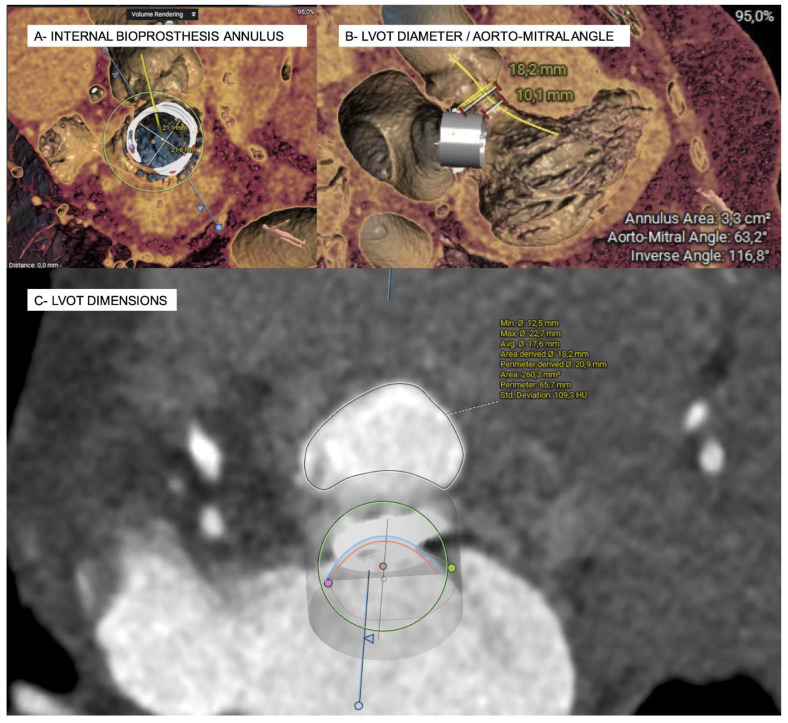
Simulation of a mitral valve-in-valve implantation and estimation of the neo-LVOT surface area on MSCT in systole using the 3 Mensio structural heart module software (Pie Medical imaging, Bilthoven, The Netherlands).

**Table 1 jcm-11-05210-t001:** Baseline characteristics.

Patient	Sex	Age (Years)	Previous Bio-Prosthesis	True Internal Diameter ^ (mm)	Internal Perimeter (mm)	Internal Area (mm^2^)	Myval Size
**1**	Female	63	Epic 29	25.8	81.9	550.8	29
**2**	Male	63	Mosaic 25	22.7	70.6	399.5	23
**3**	Male	80	Hancock-II 25	21.6	68.7	333	23
**4**	Female	65	Hancock II 25	20.7	69	390	24.5
**5**	Female	70	Epic 27	23.3	78.5	498.9	27.5
**6**	Female	57	Epic 27	23.3	72.2	416.1	24.5
**7**	Male	71	Mosaic 25	21.6	67.9	353	23
**8**	Female	71	Mosaic 25	21.7	68.1	360	23
**9**	Female	65	Hancock II 25	20.7	67.7	396	24.5
**10**	Male	63	Dokimos 27 *	24.9	80.2	506.8	27.5
**11**	Female	83	Perimount 27	23.9	75.8	444.6	23

* First case reported. ^ Internal diameter estimated as mean from perpendicular diameters.

**Table 2 jcm-11-05210-t002:** Procedural aspects, and pre- and postprocedural echocardiographic parameters.

Patient	Primary Access	Myval Size	Pre -Procedure Mean Gradient (mmHg)	Pre -Procedure Peak Gradient (mmHg)	Post- Procedure Mean Gradient (mmHg)	Post- Procedure Peak Gradient (mmHg)	Adverse Events
**1**	Femoral	29	17	34	4	6	No
**2**	Femoral	23	13	24	7	13	No
**3**	Femoral	23	17	32	4	9	No
**4**	Femoral	24.5	12	18	5	8	No
**5**	Femoral	27.5	17	23	2	-	No
**6**	Femoral	24.5	15	30	-	-	No
**7**	Femoral	23	16	29	2	6	No
**8**	Femoral	23	16	29	2	6	No
**9**	Femoral	24.5	16	29	2	5	Vascular major bleeding
**10**	Femoral	27.5	13	22	3	5	No
**11**	Femoral	23	10	-	7.6	-	No

**Table 3 jcm-11-05210-t003:** Suggested valve sizing of Myval device for mitral valve-in-valve procedures.

MITRAL VALVE-IN-VALVE SIZING CHART			
Prosthesis Size	True ID	Current THV (Sapien 3)	Myval Size
Biocor-Epic/Epic Plus			
25	21	23	24.5
27	23	26	27.5
29	25	26/29	27.5/29
31	28	29	30.5/32
33	28.5	29	30.5/32
CE Magna 7300 TFX			
25	24	26	27.5
27	26	29	29/30.5
29	28	29	30.5/32
31	28.5	29	30.5/32
33	28.5	29	30.5/32
CE Perimount 6900p-6900PTFX			
25	23	26	27.5
27	25	26/29	27.5/29
29	27	27	30.5
31	28.5	29	30.5/32
33	28.5	29	30.5/32
CE SAV Porcine 6650			
25	22.5	26	27.5
27	24	26	27.5
29	25	26/29	27.5/29
31	27	29	30.5
33	28	29	30.5/32
CE Standard Porcine 6625			
25	21	23	24.5
27	23	26	27.5
29	25	26/29	27.5/29
31	27	29	30.5
33	28	29	30.5/32
35	30.5	29 (with caution)	32
Hancock II			
25	20.5	23	23/24.5
27	22	23/26	24.5/26
29	24	26	27.5
31	26	29	29/30.5
33	28	29	30.5/32
Mosaic			
25	20.5	23	23/24.5
27	22	23/26	24.5/26
29	24	26	27.5
31	26	29	29/30.5
33	28	29	30.5/32
Pericarbon			
19	15	-	20
21	17	20	21.5
23	19	20/23	23/24.5
25	21	23	24.5
27	23	26	27.5
29	25	26/29	27.5/29
31	27	29	30.5
33	29	29	32

## Data Availability

Data are available for other investigators under request.

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
