# Peer review of "Transcatheter Mitral Valve-in-Valve Implantation with the Balloon-Expandable Myval Device"

_jcm, 2022, doi:10.3390/jcm11175210_

Round 1
Reviewer 1 Report
The authors present a retrospective case series of use of the balloon-expandable MyVal valve for mitral valve-in-valve procedures in 11 patients from 5 centers. In all cases the MyVal valve was implanted successfully without severe complications. Furthermore, the authors present a suggestion of a sizing chart for mitral valve-in-valve procedures with the MyVal valve, since this valve is not represented in the commonly used “Mitral Valve-in-Valve app”, however might be a relevant alternative to the in most cases used balloon-expandable Edwards Sapien 3 valve. The article is presented in a clear manner with the current scientific background. To my knowledge, this is the first case series of the MyVal valve used for mitral valve-in-valve procedures, however, some further clarifications are needed:
The authors describe statistical methods, however, none of these are used in the study, since there was no statistical analysis. Please correct the section "data collection and statistical analysis"
The authors mention, that in addition to the true internal diameter of the mitral bioprosthesis perimeter and area had been measured. These should also be displayed in the first table as well as the exact description of the true inner diameter (average, perimeter derived or area derived?). Table 1 and 2 do not seem to be consistent, since in table 2 the order of patients is not ascending. The tables should be reorganzied to clearly display clearly the measured inner diameter, the measured area and the then chosen size of the MyVal valve in a single table.
Patient 2, 7 ad 8 had the Mosaic 25 mm valve, however, there is a relevant difference between the true internal diameter (twice 20.5 mm, once 22.5 mm). That seems to be too much of a difference in size, please elaborate. Furthermore, were the measurements of the true internal diameter rounded up or down, since there are only .0 or .5 mm measurements.
Even though there has been no case report of use of the Myval valve in a mitral v-i-v procedure, there have been reports of two cases of the Myval valve in tricuspid v-i-v and v-i-ring procedures and one case of use of the Myval valve in a mitral v-i-ring procedure. The authors should consider citing these.
1) Two case reports of transcatheter valve-in-valve implantation of Sapien 3 and MyVal in degenerated biological tricuspid prosthesis valves. Jensen RV et al, Eur Heart J CaseRep. 2022 Mar 23;6(4) PMID:35434509
2) Transcatheter treatment tricuspid regurgitation by valve-in-ring implantation with a novel balloon-expandable Myval® THV. Ayhan H, Duran Karaduman B, KeleÅŸ T, Bozkurt E.Kardiol Pol. 2022;80(3):363-364. doi: 10.33963/KP.a2022.0013. Epub 2022 Jan 18.PMID: 35040116 3)
3) Tip-to-base LAMPOON to prevent left ventricular outflow tract obstruction in a valve-in-ring transcatheter mitral valve replacement: First LAMPOON procedure in Turkey and first LAMPOON case for transseptal Myval™ implantation. Kilic T et al, Anatol J Cardiol. 2021 May;25(5):363-367
Author Response
Answers to Reviewer’s comments #1:
- The authors present a retrospective case series of use of the balloon-expandable MyVal valve for mitral valve-in-valve procedures in 11 patients from 5 centers. In all cases the MyVal valve was implanted successfully without severe complications. Furthermore, the authors present a suggestion of a sizing chart for mitral valve-in-valve procedures with the MyVal valve, since this valve is not represented in the commonly used “Mitral Valve-in-Valve app”, however might be a relevant alternative to the in most cases used balloon-expandable Edwards Sapien 3 valve.The article is presented in a clear manner with the current scientific background. To my knowledge, this is the first case series of the MyVal valve used for mitral valve-in-valve procedures, however, some further clarifications are needed:
We thank the reviewer for the useful comments that helped us to improve our manuscript.
- The authors describe statistical methods, however, none of these are used in the study, since there was no statistical analysis. Please correct the section "data collection and statistical analysis"
We have corrected the methods accordingly.
- The authors mention, that in addition to the true internal diameter of the mitral bioprosthesis perimeter and area had been measured. These should also be displayed in the first table as well as the exact description of the true inner diameter (average, perimeter derived or area derived?). Table 1 and 2 do not seem to be consistent, since in table 2 the order of patients is not ascending. The tables should be reorganzied to clearly display clearly the measured inner diameter, the measured area and the then chosen size of the MyVal valve in a single table.
Definition of calculation for internal diameter is reflected at table foot (mean from perpendicular diameters).
Measured internal perimeter and area are now displayed in Table 1.
Valve sizing is now reported in Table 1.
- Patient 2, 7 ad 8 had the Mosaic 25 mm valve, however, there is a relevant difference between the true internal diameter (twice 20.5 mm, once 22.5 mm). That seems to be too much of a difference in size, please elaborate. Furthermore, were the measurements of the true internal diameter rounded up or down, since there are only .0 or .5 mm measurements.
The cases have been re-analysed and small differences founded in internal measurements. The exact values obtained have been reported instead of rounded up. Finally, the Myval size for patient 8 has been corrected (all 3 patients harbouring a Mosaic 25 valve received a Myval 23mm device).
- Even though there has been no case report of use of the Myval valve in a mitral v-i-v procedure, there have been reports of two cases of the Myval valve in tricuspid v-i-v and v-i-ring procedures and one case of use of the Myval valve in a mitral v-i-ring procedure. The authors should consider citing these.
1) Two case reports of transcatheter valve-in-valve implantation of Sapien 3 and MyVal in degenerated biological tricuspid prosthesis valves. Jensen RV et al, Eur Heart J CaseRep. 2022 Mar 23;6(4) PMID:35434509
2) Transcatheter treatment tricuspid regurgitation by valve-in-ring implantation with a novel balloon-expandable Myval® THV. Ayhan H, Duran Karaduman B, KeleÅŸ T, Bozkurt E.Kardiol Pol. 2022;80(3):363-364. doi: 10.33963/KP.a2022.0013. Epub 2022 Jan 18.PMID: 35040116 3)
3) Tip-to-base LAMPOON to prevent left ventricular outflow tract obstruction in a valve-in-ring transcatheter mitral valve replacement: First LAMPOON procedure in Turkey and first LAMPOON case for transseptal Myval™ implantation. Kilic T et al, Anatol J Cardiol. 2021 May;25(5):363-367
All the references suggested have been added. Given that the introduction has been modified, the references have varied along the manuscript.

Reviewer 2 Report
This is the first multicenter report of TMVR using the Myval device in degenerated mitral bioprostheses.
Even though the total number of patients (n=11) is small, this first series allows a glimpse into what is feasible with the Myval system. Immediate outcomes were compelling with very low mean gradients and absence of LVOT compromise. The availability of intermediate sizes of the Myval system could play a role in terms of favorable outcomes.
I have following comments:
- The Introduction section is well written, but should be shortened, as the multitude of information is distracting. Especially the first paragraph would be considered to be deleted completely.
- Overall, the data provided that is of relevance could be reduced to a minimum. Hence, the authors and the editorial board may consider to change the format to a short communication.
Author Response
Answers to Reviewer’s comments #2:
- This is the first multicenter report of TMVR using the Myval device in degenerated mitral bioprostheses.
Even though the total number of patients (n=11) is small, this first series allows a glimpse into what is feasible with the Myval system. Immediate outcomes were compelling with very low mean gradients and absence of LVOT compromise. The availability of intermediate sizes of the Myval system could play a role in terms of favorable outcomes.
We thank the reviewer for the comments that helped us to improve the manuscript.
I have following comments:
- The Introduction section is well written, but should be shortened, as the multitude of information is distracting. Especially the first paragraph would be considered to be deleted completely. Overall, the data provided that is of relevance could be reduced to a minimum. Hence, the authors and the editorial board may consider to change the format to a short communication.
The introduction has been significantly shortened and all the first paragraph removed as suggested.
The editorial board has not requested further shortening or format modification of the manuscript.

Round 2
Reviewer 2 Report
The comments of both reviewers were addressed appropriately. The Introduction section has been shortened considerably. I have no further comments.